# Altered Gastric Microbiota and Inflammatory Cytokine Responses in Patients with *Helicobacter pylori*-Negative Gastric Cancer

**DOI:** 10.3390/nu14234981

**Published:** 2022-11-23

**Authors:** Han-Na Kim, Min-Jeong Kim, Jonathan P. Jacobs, Hyo-Joon Yang

**Affiliations:** 1Medical Research Institute, Kangbuk Samsung Hospital, Sungkyunkwan University School of Medicine, Seoul 03181, Republic of Korea; 2Department of Clinical Research Design and Evaluation, SAIHST, Sungkyunkwan University, Seoul 06355, Republic of Korea; 3Vatche and Tamar Manoukian Division of Digestive Diseases, Department of Medicine, David Geffen School of Medicine at UCLA, Los Angeles, CA 90095, USA; 4Division of Gastroenterology, Hepatology and Parenteral Nutrition, Veterans Administration Greater Los Angeles Healthcare System, Los Angeles, CA 90073, USA; 5Division of Gastroenterology, Department of Internal Medicine and Gastrointestinal Cancer Center, Kangbuk Samsung Hospital, Sungkyunkwan University School of Medicine, Seoul 03181, Republic of Korea

**Keywords:** stomach neoplasms, gastric microbiota, 16S rRNA, cytokines, interleukin-1beta

## Abstract

The role of the gastric mucosal microbiome in *Helicobacter pylori*-negative gastric cancer (GC) remains unclear. Therefore, we aimed to characterize the microbial alterations and host inflammatory cytokine responses in *H. pylori*-negative GC. Gastric mucosal samples were obtained from 137 *H. pylori*-negative patients with GC (*n* = 45) and controls (chronic gastritis or intestinal metaplasia, *n* = 92). We performed 16S rRNA gene sequencing (*n* = 67), a quantitative reverse transcription-polymerase chain reaction to determine the relative mRNA expression levels of *TNF* (tumor necrosis factor), *IL1B* (interleukin 1 beta), *IL6* (interleukin 6), *CXCL8* (C-X-C motif chemokine ligand 8), *IL10* (interleukin 10), *IL17A* (interleukin 17A), *TGFB1* (transforming growth factor beta 1) (*n* = 113), and the correlation analysis between sequencing and expression data (*n* = 47). Gastric mucosal microbiota in patients with GC showed reduced diversity and a significantly different composition compared to that of the controls. *Lacticaseibacillus* was significantly enriched, while *Haemophilus* and *Campylobacter* were depleted in the cancer group compared to the control group. These taxa could distinguish the two groups in a random forest algorithm. Moreover, the combined relative abundance of these taxa, a GC microbiome index, significantly correlated with gastric mucosal *IL1B* expression, which was elevated in the cancer group. Overall, altered gastric mucosal microbiota was found to be associated with increased mucosal *IL1B* expression in *H. pylori*-negative GC.

## 1. Introduction

Gastric cancer (GC) remains a major disease burden that was responsible for more than 1 million new patients and 769,000 deaths globally in 2020 [1]. GC is an inflammation-associated cancer linked to a pathogenic bacterium [2]. Chronic *Helicobacter pylori* infection induces chronic gastritis (CG), develops pre-malignant lesions including mucosal atrophy and intestinal metaplasia (IM), and eventually causes GC [3]. *H. pylori* colonization is sometimes lost during the later stages of this process, but the risk of developing GC persists in these *H. pylori*-negative patients [4]. Although *H. pylori* eradication reduces the risk of GC [5,6,7], some patients continue to develop GC. Hence, gastric microbiota other than *H. pylori* may play an important role in GC occurrence.

Recent studies have reported that altered gastric microbiota are associated with GC using 16S rRNA gene sequencing [8,9,10,11,12,13]. Gastric microbial diversity is lower in patients with GC than that in controls [8,9,10]. Several bacterial taxa are enriched or depleted in the gastric microbiota [8,9,10]. However, because *H. pylori* has a significant impact on the gastric microbiome composition, a separate analysis may be necessary to evaluate the gastric microbiome alterations in *H. pylori*-negative GC.

Chronic *H. pylori* infection induces excessive and chronic production of pro-inflammatory cytokines, such as IL1B (interleukin 1 beta), IL6 (interleukin 6), CXCL8 (C-X-C motif chemokine ligand 8), which are thought to promote the development and progression of GC [14]. Similarly, in *H. pylori*-negative patients, other microbiota may induce pro-inflammatory cytokines and possibly promote GC development. However, our understanding of the interactions between gastric microbiota and ongoing inflammation in *H. pylori*-negative patients is still limited.

Therefore, in this study, we aimed to characterize the microbial alterations in *H. pylori*-negative patients with GC compared to those in patients with CG and IM. We also evaluated the correlation between microbial alterations and the host inflammatory cytokine responses in these patients.

## 2. Materials and Methods

### 2.1. Study Participants and Sample Collection

In this study, we included patients with CG, IM, or GC aged between 19 and 75 years who underwent upper endoscopy between April, 2020–April, 2021 at Kangbuk Samsung Hospital, Seoul, Korea. We excluded (1) patients who took proton-pump inhibitors, H_2_ receptor antagonists, mucoprotective agents, antacids, probiotics, or antibiotics within 1 month, (2) patients who underwent *H. pylori* eradication within 1 year to exclude the residual effect of recent *H. pylori* infection, and (3) patients who underwent gastrectomy.

Demographic data, including age, sex, and body mass index (BMI), were collected from the patients. During endoscopy, gastric mucosal tissues were taken from the greater curvature side of the mid-antrum of the stomach for microbiome and RNA analyses, immediately frozen at −20 °C, and stored at −70 °C for 6 h. To avoid contamination, a disinfected endoscope was used according to the standard sterilization protocol [15,16]. Patients fasted overnight and rinsed their mouths before endoscopy. An experienced gastrointestinal endoscopist performed the procedure using a single-channel endoscope (GIF-H290; Olympus Optical, Tokyo, Japan). Endoscopic biopsy samples were obtained before any fluid was suctioned through the scope. Endoscopic procedure included detailed observation of esophageal, gastric, and duodenal lesions. Gastric mucosal atrophy was evaluated endoscopically. Gastric IM was evaluated by histological evaluation using endoscopic biopsy tissues taken from both the lesser curvature side of the mid-antrum and the lesser curvature side of the mid-body of the stomach. *H. pylori* infection was evaluated by histological evaluation with modified Giemsa staining and a rapid urease test, and was considered negative when both tests were negative. This study was approved by the Institutional Review Board of Kangbuk Samsung Hospital (KBSMC 2020-03-027). Written informed consent was obtained from all the patients prior to their participation in this study.

### 2.2. DNA Extraction from Gastric Biopsy Samples and 16S rRNA Gene Sequencing

Total DNA was extracted from gastric biopsy tissues using the DNeasy PowerSoil Kit (Qiagen, Hilden, Germany), according to the manufacturer’s instructions. V3 and V4 regions of the bacterial 16S rRNA gene were amplified using the universal primers 337F to 805R to construct sequencing libraries that underwent 2 × 300 bp paired-end sequencing on the Illumina 16S MiSeq platform (Illumina Inc., San Diego, CA, USA). A negative control was used for DNA extraction and 16S rRNA gene amplification, and no band was observed in 0.2% agarose gel after 30 cycles of amplification.

### 2.3. Sequence Data Processing

Raw sequence data were processed using Quantitative Insights Into Microbial Ecology 2 (QIIME2) 2021.4. The DADA2 plugin was used to truncate and trim the low-quality base calls of demultiplexed reads. After quality control, the reads were denoised and generated as amplicon sequence variants (ASVs). After paired-end joining and chimera removal, 3,014,910 reads (mean per sample = 44,999 ± 21,104 SD) were found in 67 samples. Each ASV was taxonomically assigned using the National Center for Biotechnology Information (NCBI) nucleotide and taxonomy databases (NCBI-RefSeq; accessed on 9 June 2021) using RESCRIPt within QIIME2.

Final sample meta-data, ASVs, and taxonomy tables were exported from QIIME2 and further processed using MetagenomeAnalyst (accessed on 10 August 2022) [17]. Prior to conducting downstream analysis, the sequences were filtered to remove the singleton reads that were unassigned at the phylum level. Finally, 1014 ASVs remained in 67 samples.

### 2.4. Microbial Profiling

For the α-diversity analysis, we rarefied the sequence depth to 6814 read counts in all samples. Sample completeness plateaued at approximately 1400 reads in the rarefaction curves (Appendix A). Alpha diversity was analyzed using the *phyloseq* package in MicrobiomeAnalyst. Alpha diversity included the total number of species (richness), abundance of the species (evenness), or measures that consider both richness and evenness. We calculated richness using the actual number of unique taxa observed in each sample (observed index) and used the Shannon index to describe both richness and evenness.

For downstream analysis, we additionally filtered rare features so that the complexity was reduced while the integrity was preserved [18]. In the MicrobiomeAnalyst data filtering step, we used the settings of “Low count filter” with a minimum count of 4 and 10% prevalence in samples, and the “percentage to remove” option under “Low variance filter” set to 10% based on the interquartile range. A total of 905 low-abundance ASVs and 11 low-variance ASVs were removed, leaving behind 98 ASVs.

The dissimilarity between the two groups was estimated using the phylogenetic and non-phylogenetic β-diversity indices. UniFrac distance was used as the phylogenetic index representing the absence or presence of ASVs for weighted UniFrac distance and the abundance of ASVs for unweighted UniFrac distance. Jaccard distance and Bray–Crutis dissimilarities were used as the non-phylogenetic indices, representing the absence or presence and abundance of ASVs, respectively. Permutational multivariate analysis of variance (PERMANOVA) was conducted with 999 random permutations to test the significance of the differences between the groups.

We also conducted hierarchical cluster analysis. Each sample began as a separate cluster, and the algorithm combined them until all samples belonged to one cluster. We used the default parameters, “Euclidean” for distance measure and “Ward” for clustering algorithm, to perform hierarchical clustering. In MicrobiomeAnalyst, the results of clustering analysis are shown as a heatmap using the hclust function in the package stat.

The differences between the two groups in the relative abundances of taxa ranging from the phylum to species levels were evaluated using the analysis of comparison of microbiomes (ANCOM) 2 (https://github.com/FrederickHuangLin/ANCOM; accessed on 23 September 2022). The significant findings in this analysis were validated using generalized linear models implemented in multivariate association with linear models (MaAsLin2). Both analyses were conducted using R (version 4.0.2; R Foundation for Statistical Computing, Vienna, Austria). The taxa-wise false discovery rate (FDR) adjusted *p* value using Benjamini-Hochberg adjustment for the multiple testing correction method was used for the ANCOM2. An ANCOM2 detection level ≥ 0.7 was considered significant as it indicates that the ratios of the taxon to at least 70% of other taxa were detected to be significantly different (FDR, *q* < 0.10) between the cancer and control groups. The significance threshold of the detection level of 0.7 was recommended by the author of ANCOM2 as a common choice [19]. For MaAsLin2 models, the control group was used as the reference group, and the FDR correction was not used because these models were used only for the validation of the significant findings in the analysis using ANCOM2. Both analyses using ANCOM2 and MaAsLin2 were adjusted for potential covariates of age, sex, and BMI.

We used linear discriminant analysis effect size (LEfSe) to identify the potential bacterial markers for GC. Only taxa with a linear discriminant analysis (LDA) score (log10) > 3 (*p* < 0.05) were considered to be significantly enriched. We ranked the bacteria related to GC according to their importance in the prediction of GC using random forest (RF), a machine learning method. The importance score was assigned by increased error when the feature was removed from the prediction model. The data were further transformed to a centered log ratio before applying the RF classification algorithm. We used a parameter of 2000 trees to predict from and seven predictors to try (mtry) with the randomness setting left in the RF algorithm within MicrobiomeAnalyst. The overall error rate and error rate for each group were calculated based on the results of each run.

### 2.5. RNA Extraction and Quantitative Reverse Transcription-Polymerase Chain Reaction (RT-qPCR)

Total RNA samples were prepared using the RNeasy Plus Micro Kit (#74034; Qiagen, Hilden, Germany). Briefly, gastric mucosa tissues were homogenized in Buffer RLT Plus using the MagNA Lyser homogenizer (Roche, Mannheim, Germany). The lysates were centrifuged at full speed for 3 min, the supernatant was carefully transferred to a gDNA Eliminator spin column, and RNA was isolated according to the manufacturer’s protocol. mRNA was reverse-transcribed using the High-Capacity RNA-to-cDNA Kit (Applied Biosystems, San Diego, CA, USA), and qPCR was performed using a specific primer and SensiFAST SYBR Lo-ROX Kit (Bioline, Taunton, MA, USA) on a LightCycler 480 System (Roche, Mannheim, Germany). Primer sequences used in this study are listed in Appendix A. The relative mRNA expression of each gene was normalized to that of the housekeeping gene (*L32*) and calculated using the 2^−ΔΔCT^ method.

### 2.6. Correlation between the Altered Gastric Microbiome and Cytokine Response

Correlations between the microbial profiles, including α-diversity indices and relative abundances of individual taxa, and the relative mRNA expression levels of pro-inflammatory cytokines, were evaluated using Pearson’s correlation test after log-transformation. To evaluate the role of key taxa in the prediction of GC, we combined the three most important taxa in the RF model into an index by adding the log-transformed relative abundance of taxa in a positive correlation with GC and subtracting those in a negative correlation. We then evaluated the correlation between this index and the relative cytokine expression. Statistical analyses of cytokine expression and correlation analyses were performed using R. The FDR-corrected *q* value was also used for the results of mRNA expression and correlation analyses.

## 3. Results

### 3.1. Study Demographics

This study included 137 *H. pylori*-negative patients, comprising 45 patients with GC (cancer group) and 92 controls (49 patients with CG and 43 with IM) (Figure 1). Patients in the cancer group were significantly older and more likely to be male and obese than those in the control group (Table 1). This study was conducted in three steps: First, 16S rRNA gene sequencing analysis was performed using 74 samples. After excluding seven samples with a relative abundance of *H. pylori* > 0.01, which was considered a false-negative result for *H. pylori* [20], 67 samples were subjected to microbial profiling. Second, 113 samples were used for RT-qPCR analysis. Finally, 47 samples with both microbiome profiles and inflammatory cytokine expression data were analyzed for their correlations.

### 3.2. Microbial Diversity Associated with GC

First, we analyzed the α-diversity to examine the differences in gastric microbial richness and evenness between the cancer and controls groups. Patients with GC had lower microbial richness than the controls, as indicated by the observed ASVs (Mann–Whitney *U* test, *p* = 0.011; Figure 2A). Considering both richness and evenness, there was no difference in the Shannon index between the two groups (*p* = 0.090; Figure 2B). These results were consistent after additional adjustments for age, sex, and BMI (analysis of covariance, *p* = 0.047 and *p* = 0.898, respectively).

When the gastric mucosal microbial communities were compared (β-diversity) using PERMANOVA, differences in the overall gastric microbiome composition between cancer and control groups were observed in the phylogenetic indices: weighted UniFrac distance (PERMANOVA, R^2^ = 0.047, *p* = 0.019) and unweighted UniFrac distance (R^2^ = 0.028, *p* = 0.047) (Figure 3A,B). Principal coordinate analysis visually showed a shift in the distribution of cancer compared to that in the control, which was consistent with significant differences revealed by PERMANOVA. No statistically significant differences were observed between the two groups using non-phylogenetic β-diversity indices, including Bray–Curtis (R^2^ = 0.025, *p* = 0.062) and Jaccard distance (R^2^ = 0.021, *p* = 0.082) (Figure 3C,D).

### 3.3. Taxonomic Alterations in GC

Using the ANCOM2 method, we identified several highly abundant taxa from the phylum to species level that differed between the control and cancer groups (Table 2). Proteobacteria and Firmicutes were the two most dominant phyla in the stomach of both groups, with an average of 76.0% (Proteobacteria 52.8% and Firmicutes 23.2%) and 84.7% (Proteobacteria 58.2% and Firmicutes 26.5%) gastric bacteria in the control and cancer groups, respectively (Appendix A). However, the abundance of the most abundant phyla was not significantly different between the cancer and control groups. We found that *Lacticaseibacillus casei* was significantly enriched (detected >0.7; MaAsLin2, log_2_ fold change [FC] = 1.017) in the cancer group, while *Haemophilus parainfluenzae*, including its higher taxonomic level, and the genus *Campylobacter* were depleted in the cancer group compared to that in the control group. Among the significant taxa, *Haemophilus parainfluenzae* was the most significantly different taxon relative to at least 90% of the other taxa (ANCOM2, detected >0.9; MaAsLin2, log_2_ FC = −1.488). The statistical significance of the taxa remained after adjusting for age, sex, and BMI. All ANCOM2-detected differentially abundant bacteria between the two groups were also confirmed to be significant using the MaAsLin2 method (*p* < 0.05) with or without adjusting for covariates, except *Haemophilus parainfluenza*, which was marginally significant (*p* = 0.052) in MaAsLin2 after covariate adjustments. *Lacticaseibacillus* and *L. casei* were more significant (ANCOM2, detected >0.7; MaAsLin2, *p* = 0.006) and showed higher effect sizes (MaAsLin2, log_2_ FC = 1.405) after adjusting for the covariates. We also observed a higher abundance of *Lacticaseibacillus* and a lower abundance of *Campylobacter* and *Haemophilus* in samples with GC compared to the control samples in the hierarchical clustering heatmap, although the heatmaps were not distinctly clustered by the two groups (Figure 4).

### 3.4. Identification of Key Gastric Microbiota Associated with GC via LEfSe and RF Analyses

To assess the robustness of the taxonomic differences between the control and cancer groups, we performed an LEfSe analysis. LEfSe combines statistical significance and effect size, or LDA score, indicating how well the abundance of each bacteria distinguishes the groups. This method confirmed the significance (|LDA score| > 4, *p* < 0.05) of the genera *Lacticaseibacillus*, *Haemophilus*, and *Campylobacter* (Figure 5A), and the species *Haemophilus parainfluenzae* and *L. casei* (Figure 5B). In addition, this analysis highlighted *Brevundimonas* with *Lacticaseibacillus* in cancer and *Porphyromonas* and *Schaalia* with *Haemophilus* and *Campylobacter* in controls as important taxa that distinguished the two groups (Figure 5A). At the species level, *Brevundimonas vesicularis* in the cancer group and *Porphyromonas pasteri*, *Campylobacteraceae concisus*, *Haemophilus haemolyticus*, and *Schaalia odontolytica* in the control group were found to be significant in the LEfSe analysis (|LDA score| > 3, *p* < 0.05) (Figure 5B).

Although numerous bacteria were differentially abundant between the control and cancer groups, we wanted to identify the key gastric microbiota that could best discriminate between the cancer and control groups. To do this, we used an RF algorithm to classify the samples into control or cancer groups using the genus abundance data. Important bacteria were then identified based on the extent to which the RF classification accuracy decreased when the bacteria were removed from the feature set. *Haemophilus*, *Campylobacter*, and *Lacticaseibacillus*, and *Haemophilus parainfluenzae*, *Veillonella atypica,* and *Bifidobacterium animalis* were identified as the top three important features at the genus (Figure 5C) and species levels (Figure 5D), respectively, to distinguish the control and cancer groups. The out-of-bag error was 0.254, 0.284 and the class error was 0.065, 0.065 for the control group and 0.667, 0.762 for the cancer group at the genus and species levels, respectively (Appendix A), indicating that bacterial features could distinguish the control group from the cancer group, but not vice versa.

### 3.5. Altered Mucosal Inflammatory Cytokine Expression in GC

In RT-qPCR analysis, the relative mRNA expression level of *IL1B* in the gastric mucosa was significantly higher in the cancer group than that in the control group (Mann–Whitney test, *p* = 0.030) (Figure 6A). However, this was not significant after the correction for multiple comparison (*q* = 0.210). Although the mean expression levels of *IL6, CXCL8, IL10*, and *IL17A* were higher in the cancer group than those in the control group, the difference was not statistically significant. The mRNA expression of *IL1B* also showed a significantly increasing trend in CG, IM, and GC in the three-group comparison even after the FDR correction (linear regression analysis, *p*-for-trend = 0.009, *q* = 0.063) (Figure 6B).

### 3.6. GC Microbiome Index Associated with the Inflammatory Cytokine Response

We analyzed the correlation between the microbial profile associated with GC and pro-inflammatory cytokine expression. The patients included in this analysis showed similar baseline characteristics (Appendix A), relative abundance of the differentially abundant taxa (Appendix A), and cytokine expression profile (Appendix A) compared to the overall population. After excluding outliers with values outside the mean ± 2 standard deviations, the observed index that was reduced in GC was negatively correlated with the mRNA expression of *IL1B* (*q* = 0.033, r = −0.3260) (Figure 7A). However, when we further excluded two more highest value points of the observed index, the correlation became no longer significant (*q* = 0.214, r = −0.2267), suggesting that this correlation may have been driven by several high values of the α-diversity. Among the differentially associated taxa between GC and control groups, the relative abundance of *Campylobacter* was negatively correlated with *IL1B* (*q* = 0.008, r = −0.4328) (Appendix A). However, because *Campylobacter* was absent in many samples, this correlation might have driven by several samples with high abundance of this taxon. As the genera *Campylobacter*, *Lacticaseibacillus*, and *Haemophilus* were the three most important taxa in the RF model, we combined the relative abundance of these taxa to create a GC microbiome index. The GC microbiome index was significantly positively correlated with *IL1B* mRNA expression (*q* = 0.031, r = 0.3185).

## 4. Discussion

Here, we evaluated the gastric microbiome associated with GC in *H. pylori*-negative patients. The richness of the gastric microbiota was significantly lower in patients with GC than those in the controls. The gastric microbiota in GC was significantly different from that in CG and IM in terms of phylogenetic *β*-diversity. The gastric microbiome in cancer was characterized by the enrichment of *Lacticaseibacillus* and the depletion of *Campylobacter* and *Haemophilus*, from which we derived the GC microbiome index. Importantly, we showed that gastric microbial diversity, specific taxon composition including *Campylobacter*, and the GC microbiome index were significantly correlated with gastric mucosal *IL1B* mRNA expression, which was elevated in patients with GC.

Previous studies on the gastric microbiota in patients with GC mostly reported significantly different microbial compositions in these patients compared to the controls, including patients with CG and IM [8,10,13,21,22,23]. However, mixed results were observed regarding changes in bacterial α-diversity in GC. Many studies reported a decrease in the overall diversity in GC [8,9,13,21,23], while some others reported increased diversity [10,22,24]. In addition to differences in ethnicity, diet, and technical factors, such as 16S rRNA gene targets and sequencing platforms, these conflicting results may be attributable to the mixed population of *H. pylori*-positive and -negative patients. Loss of *H. pylori* and impaired acid secretion in the later stages of gastric carcinogenesis may lead to increased bacterial colonization of the stomach [25]. *H. pylori* eradication also results in increased bacterial diversity [26]. In our study, we strictly excluded *H. pylori*-positive patients and found that the overall diversity decreased in *H. pylori*-negative GC than that in *H. pylori*-negative CG and IM. These results suggest that gastric microbial diversity may decrease during the development of GC, if changes following the loss or eradication of *H. pylori* are excluded. Taken together with increased mucosal *IL1B* mRNA expression in our study, these changes can be explained by the ongoing inflammation, which produces an inhospitable environment for most microbes [25].

Lactic acid bacteria (LAB) have been consistently reported to increase in relative abundance in GC. Most studies have reported an increased relative abundance of *Lactobacillus* in patients with GC [8,9,13,21,22,23,24]. Increased abundances of other LAB, including *Streptococcus* [8,9,24], *Bifidobacterium*, and *Lactococcus* [22], have also been reported. In our study, the relative abundance of *L. casei*, a LAB, was increased in GC. LAB also have anti-inflammatory and anticancer effects [27]. *L. casei*, as a probiotic, exerts anticancer effects on GC cells [28]. However, LAB overgrowth can reduce nitrate to nitrite, leading to the formation of N-nitroso compounds [29]. This is supported by a study by Ferreira et al. [21] which reported an increased nitrosating function and increased abundance of *Lactobacillus* in GC. LAB can also potently induce reactive oxygen species, which are involved in the development of GC [30]. In a previous study, insulin-gastrin transgenic mice colonized with *Lactobacillus murinus* ASF361, *Clostridium* sp. ASF356, and *Bacteroides sp.* ASF519, and developed gastric intraepithelial neoplasia with strong upregulation of oncogenes and pro-inflammatory genes [31]. These findings support the positive correlation between the relative abundance of *Lacticaseibacillus* and mucosal *IL1B* mRNA expression observed in our study. Therefore, overrepresentation of *Lacticaseibacillus* may contribute to the development of GC by inducing chronic inflammation in *H. pylori*-negative patients.

In our study, the relative abundances of *Haemophilus parainfluenzae* and *Campylobacter* were reduced in GC. They are members of the normal oral microflora [32,33]. In addition, we found depletion of other oral bacteria, such as *S. odontolytica* and *Prevotella salivae*, in GC via RF analysis. In a previous study, *Haemophilus* was found to be reduced in the saliva microflora of patients with GC [32]. *Haemophilus parainfluenzae* was increased in the gastric microbiome of patients with intraepithelial neoplasia but decreased in patients with GC [9]. As *Haemophilus* is a nitrate-reducing bacterium, the decreased abundance of this taxon may contribute to the chronic inflammatory process [34]. This is consistent with our finding of a negative correlation between *Haemophilus* abundance and *IL1B* mRNA levels. In contrast, *Campylobacter*, particularly *C. concisus*, has been reported to be associated with inflammatory bowel disease [35]. Notably, *C. concisus* abundance was increased in Barrett’s esophagus but not in esophageal cancer [36]. Because it induced the expression of the inflammatory cytokine *IL18* in that study, the reduced abundance of *Campylobacter* in our study may be the result, rather than the cause, of increased *IL1B* mRNA expression. This seems similar to the situation in which the relative abundance of *H. pylori* is reduced or even lost during the final stage of gastric carcinogenesis [21,24]. One study reported increased abundances of *Haemophilus* and *Campylobacter* in GC, but this may be due to their comparison to functional dyspepsia as a control group rather than a group including IM [22].

In our study, the most differentially abundant genera between cancer and control groups were validated using various analysis methods, including ANCOM2, MaAsLin2, and LEfSe analyses. In addition, the integration of these genera into an RF classifier allowed us to discriminate GC from the controls. We derived a novel GC microbiome index that significantly correlated with gastric mucosal *IL1B* expression. Our results suggest that microbial changes at the community level, rather than individual taxa, may contribute to the persistent chronic inflammation process, and through this process, may increase the risk of GC development in *H. pylori*-negative patients. The roles of individual taxa can vary. Increased *Lacticaseibacillus* and reduced *Haemophilus* abundances may induce ongoing inflammation, while reduced *Campylobacter* abundance may be a result of this ongoing inflammation.

The novelty of our study is that we identified gastric microbiota associated with GC in *H. pylori*-negative patients, which has growing importance. Previous studies have focused on the synergistic role of *H. pylori* and other gastric microbiota in GC [8,9,10,21]. Sung et al. reported gastric microbiota associated with the gastric premalignant lesion after *H. pylori* eradication [26]. We adopted strict criteria for classifying *H. pylori*-negative cases using 16S rRNA gene sequencing data, which allowed us to evaluate the gastric microbial community, excluding *H. pylori* [20]. Another novelty is that we investigated the gastric mucosal inflammatory cytokine response associated with differentially abundant gastric microbiota in GC. Recently, several studies reported the inflammatory cytokine response to the gastrointestinal microbiota in healthy control or patients with Parkinson’s disease [37,38]. The role of IL1B, IL6, and CXCL8 has been noted in the association between chronic *H. pylori* infection and the development of GC [14]. To our knowledge, this is the first study that evaluated the gastric microbiota and inflammatory cytokine response in association with *H. pylori*-negative GC. However, this study has some limitations. First, because of the strict criteria, the sample size was modest, which limited the statistical power to identify differentially abundant taxa between the cancer and control groups. Second, we did not measure cytokine levels from gastric juice or plasma. However, the measurement of mRNA expression levels of cytokine genes from gastric mucosal samples may also provide meaningful results [39,40]. Third, our findings were not validated in different ethnic populations. Although we validated our findings with this cohort using multiple analytical tools, further validation studies with replication cohorts are required.

In conclusion, this study revealed the presence of altered gastric mucosal microbiota in *H. pylori*-negative GC. Increased *Lacticaseibacillus* and reduced *Haemophilus* and *Campylobacter* abundances associated with increased mucosal *IL1B* expression levels may contribute to the development of *H. pylori*-negative GC.

## Figures and Tables

**Figure 1 nutrients-14-04981-f001:**
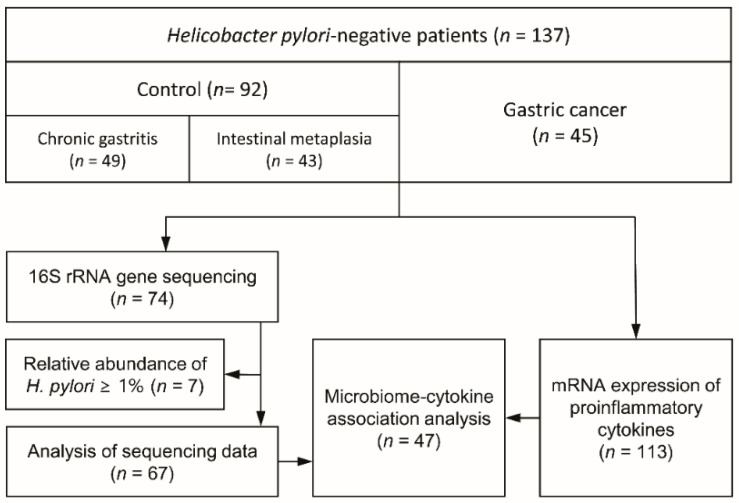
Study flow.

**Figure 2 nutrients-14-04981-f002:**
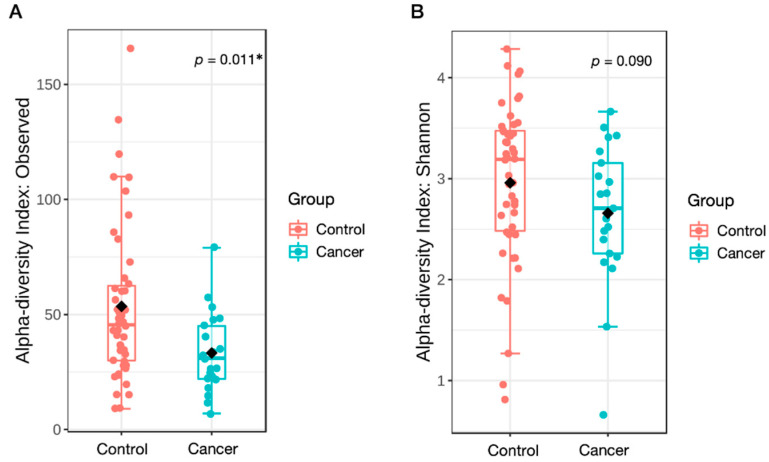
Reduced α-diversity in the gastric microbiota of patients with cancer than that in the controls. Boxplots represent the α-diversity measured using (**A**) observed amplicon sequence variants (ASVs) and (**B**) Shannon index at ASV level. Mann–Whitney test was used to compare the microbial diversity between the two groups. * *p* < 0.05. The median line, box, and whiskers indicate the median value, 25th and 75th percentiles, and 1.5 interquartile ranges, respectively.

**Figure 3 nutrients-14-04981-f003:**
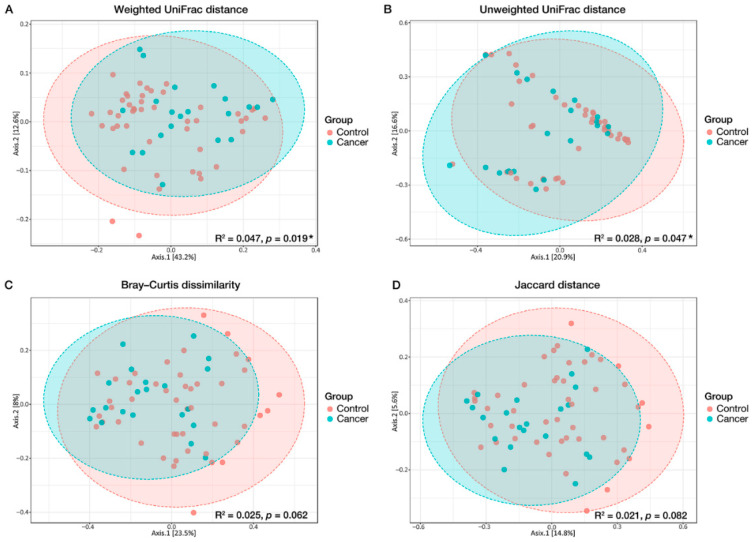
Different structures of the overall gastric microbiome composition (β-diversity) between cancer and control groups in principal coordinate analysis. (**A**) Weighted UniFrac distance, (**B**) unweighted UniFrac distance, (**C**) Bray–Curtis dissimilarity, and (**D**) Jaccard distance. The effect size and significance were assessed using permutation multivariate analysis of variance. Ellipses were drawn based on the 95% confidence interval for each group. * *p* < 0.05.

**Figure 4 nutrients-14-04981-f004:**
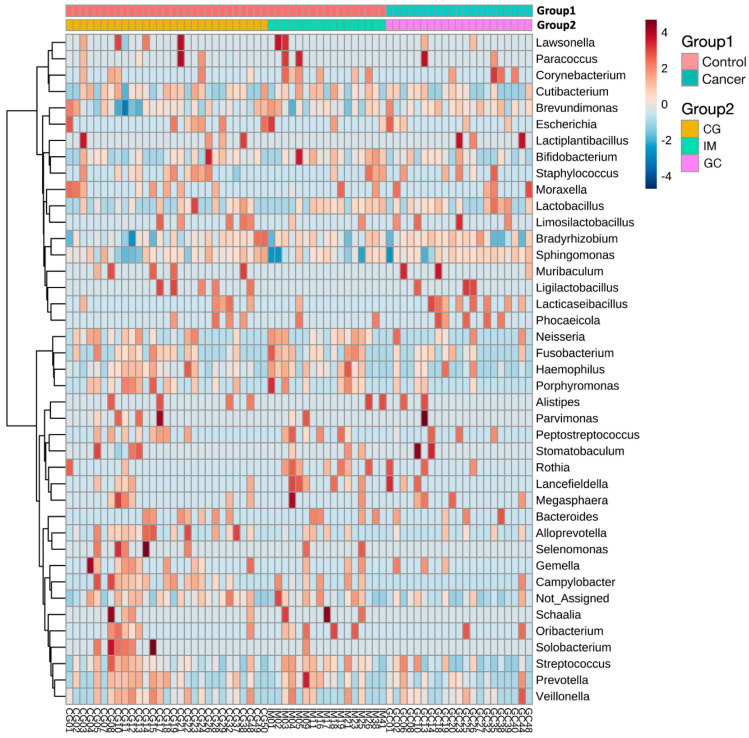
Hierarchical clustering heatmap for relative abundance of gastric microbiota between cancer and control groups (Group 1) at the genus level. Two-colored bars on the top of the figure show the grouping category. Every sample (control group = red, cancer group = green in Group 1) is represented by its own column. Scaling is by relative abundances of each taxon from low (light blue) to high (dark red) across all samples. Clustering was performed using Euclidean distance measure and Ward’s linkage clustering algorithm at the genus level. Individual patient identifiers are listed followed by CG, IM, or GC (Group 2), indicating chronic gastritis, intestinal metaplasia, or gastric cancer, respectively.

**Figure 5 nutrients-14-04981-f005:**
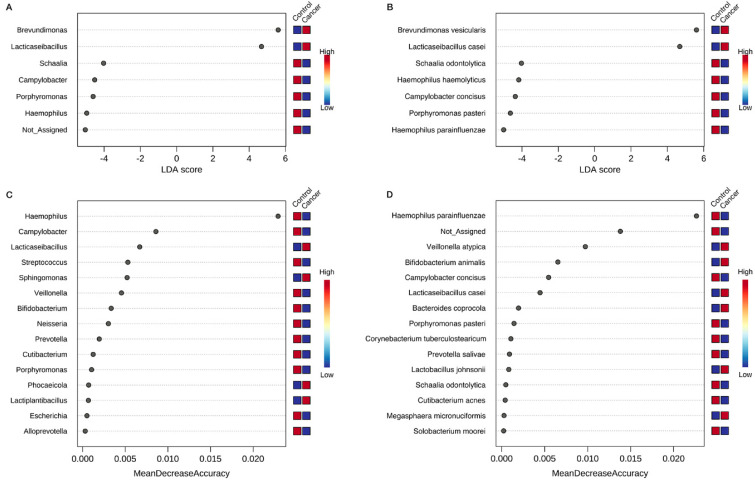
Taxonomic differences between control and cancer groups in linear discriminant analysis effect size (LEfSe) and random forest (RF) analyses. Linear discriminant analysis (LDA) score of differentially expressed bacteria obtained from LEfSe analysis of gastric microbiota in the control and cancer groups at the (**A**) genus and (**B**) species levels. Taxa with *p* < 0.05 from the Kruskal–Wallis test and LDA score > 3 are shown. In RF analysis, (**C**) the top 15 genera and (**D**) species with the highest discriminatory power between the control and cancer groups are listed. In both LEfSe and RF analyses, red fields indicate higher abundance and blue fields indicate lower abundance in the indicated group compared to the other group.

**Figure 6 nutrients-14-04981-f006:**
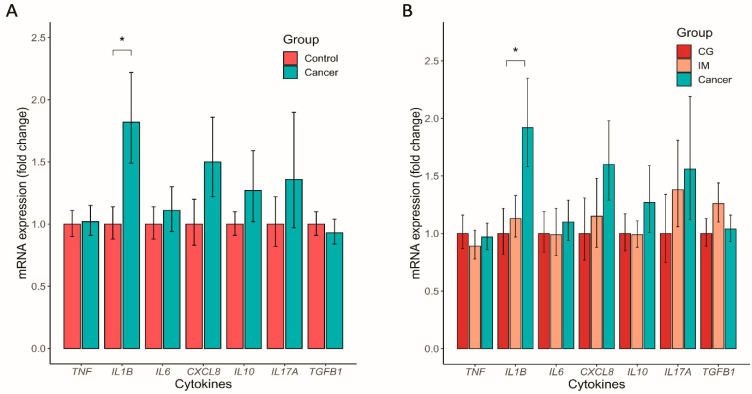
Increased relative expression levels of *IL1B* mRNA in the gastric mucosa of patients with cancer compared to those in the controls. Levels of relative mRNA expression of *TNF, IL1B, IL6, CXCL8, IL10, IL17A,* and *TGFB1* were compared (**A**) between the control and cancer groups and (**B**) between patients with chronic gastritis (CG), intestinal metaplasia (IM), and gastric cancer (GC). Levels were presented as fold change compared to the reference group (control and CG, respectively) and standard errors. * *p* < 0.05.

**Figure 7 nutrients-14-04981-f007:**
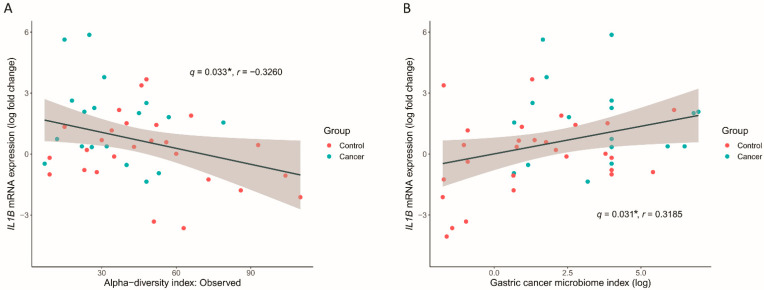
Significant correlation between the microbial profiles and *IL1B* mRNA expression levels. (**A**) Correlation of α-diversity index, observed index, with relative *IL1B* mRNA expression levels, were evaluated using Pearson’s correlation of log transformed values. (**B**) Relative abundance of *Campylobacter, Lacticaseibacillus,* and *Haemophilus* were combined to derive a gastric cancer microbiome index. Correlation of this index with the *IL1B* mRNA expression was assessed. * *q* < 0.1.

**Table 1 nutrients-14-04981-t001:** Demographics of the patients included in this study.

	Gastric Cancer (*n* = 45)	Control (*n* = 92)	*p* Value
Age, years, mean ± SD	62.9 ± 10.2	50.7 ± 13.6	<0.001
Sex, *n* (%)			0.048
Female	14 (31.1)	45 (48.9)	
Male	31 (68.9)	47 (51.1)	
Body mass index, *n* (%)			0.016
<25 kg/m^2^	24 (53.3)	67 (73.6)	
≥25 kg/m^2^	21 (46.7)	24 (26.4)	
Gastric mucosal atrophy, *n* (%)			<0.001
Absent	2 (4.4)	44 (47.8)	
Present	43 (95.6)	48 (52.2)	
Intestinal metaplasia, *n* (%)			<0.001
Absent	4 (8.9)	49 (53.3)	
Present	41 (91.1)	43 (46.7)	

Student’s *t*-test and chi-square test were used to compare the two groups. SD, standard deviation.

**Table 2 nutrients-14-04981-t002:** Differentially abundant taxa between the control and gastric cancer groups.

Taxonomic Assignment ^a^	Relative Abundance (Mean%)		ANCOM2 ^b^ (Unadjusted)	ANCOM2 ^b^ (Adjusted for Age, Sex, and BMI)	MaAsLin2 (Unadjusted)		MaAsLin2 (Adjusted for Age, Sex, and BMI)	
	Control	Cancer	Detection level	Detection level	log_2_FC ^c^	*p* ^d^	log_2_FC ^c^	*p* ^d^
p_Proteobacteria;								
o_Gammaproteobacteria;								
c_Pasteurellales;	6.29%	4.45%	0.8	0.8	−2.283	0.006	−1.862	0.052
f_Pasteurellaceae;	6.29%	4.45%	0.8	0.8	−2.283	0.006	−1.862	0.052
g_*Haemophilus;*	6.29%	4.45%	0.8	0.8	−2.283	0.006	−1.862	0.052
* s_parainfluenzae;*	4.99%	2.92%	0.9	0.9	−1.876	0.005	−1.488	0.052
p_Proteobacteria;								
o_Epsilonproteobacteria;								
c_Campylobacterales;								
f_Campylobacteraceae;								
g_*Campylobacter*	1.04%	0.38%	0.7	0.6	−2.457	0.002	−1.784	0.047
p_Firmicutes;								
o_Bacilli;								
c_Lactobacillales;								
f_Lactobacillaceae;								
g_*Lacticaseibacillus;*	0.29%	1.24%	0.6	0.7	1.017	0.021	1.405	0.006
s_*casei*	0.29%	1.24%	0.7	0.8	1.017	0.021	1.405	0.006

Taxa detected to be significantly different relative to more than 70% of other taxa at each taxon level (detected >0.7) in the analysis of comparison of microbiomes. (ANCOM)-2 analysis are included in this table. ^a^ NCBI-RefSeq database was used for taxonomic assignment. ^b^ Benjamini–Hochberg multiple comparison correction method was applied (false discovery rate < 0.1). ^c^ Coefficients for log-transformed relative abundance of each taxon in linear models using MaAsLin2. The control group was set as the reference group and compared to the cancer group. ^d^ Unadjusted *p* values for the multiple comparison. The significance of the ANCOM2 model was validated using the MaAsLin2 model. ANCOM, analysis of comparison of microbiomes; MsAsLin, multivariate association with linear models; BMI, body mass index; FC, fold change.

## Data Availability

The data presented in this study are not available at this time as the data forms are part of an ongoing study.

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
