# Peer review of "Altered Gastric Microbiota and Inflammatory Cytokine Responses in Patients with Helicobacter pylori-Negative Gastric Cancer"

_nutrients, 2022, doi:10.3390/nu14234981_

Round 1

Reviewer 1 Report

Review for nutrients-2030628

The authors investigated the role of the microbiota in H.pylori negative CG patients that went on to develop GC. While the paper is well written, there were some major problems with the presentation and thus interpretation of the results. Once important changes are made and the results are still significant, the paper could be publishable.

Major comments

1.     The authors investigated several cytokines as well as several bacterial species, but no mention of multiple testing correction is made in the entire paper. This must be changed for the correlations as well as the boxplots shown in the supplementary material. Without correction, the authors cannot be confident that the results are not spurious associations.

2.     Figure 4 has very little meaning, as the samples were not allowed to cluster. There is no observable pattern in the heatmap. The authors should consider coloring the column groups by the three groups mentioned in the caption of the figure and to allow the samples to cluster to identify any observable clusters in the data.

3.     Correlation plots are important for the understanding of the paper. In Figure 7A the 4 samples on the right, i.e. the high end of diversity, are driving the correlations. It also appears to be the same 4 samples in both plots. Please show the correlations without these 4 extreme samples.

4.     Further, why the authors are showing two measures of the same diversity when it appears to be just a different unit is a mystery. The authors should only use one of the measures and put the other into the supplementary.

5.     The first 3 panels of Figure 7B are between worrying and not true correlations, as the correlations are driven again by extreme values in those patients where the bacteria were present. Here, the authors should either not show those samples where the bacteria were absent or choose a different mode of association. A fisher test could be recommended here where the presence and absence of the bacteria are coded as 1 and 0, respectively. The only panel that showed a true correlation is the last panel of Figure 7B but since there was no multiple testing correction applied, it cannot be trusted.

Author Response

Nov 16, 2022

Prof. Dr. Maria Luz Fernandez

Prof. Dr. Lluis Serra-Majem

Editors-in-Chief, Nutrients

Dear Editors-in-Chief:

RE: [Nutrients] Manuscript ID: nutrients-2030628 - Major Revisions

We appreciate the editors and reviewers for having the chance to revise our manuscript and for the constructive suggestions regarding our manuscript (nutrients-2030628), entitled “Altered Gastric Microbiota and Inflammatory Cytokine Responses in Patients with Helicobacter pylori-Negative Gastric Cancer.” We revised the manuscript to address the comments from the reviewers and we believe that the manuscript has improved substantially in this process.

We have uploaded the revised version of the manuscript with ‘track changes’ function that reflected the modifications incorporated into the manuscript, as well as an item-by-item response to the reviewers’ comments that details the changes introduced in response to the comments.

We are pleased to submit our revised version to be considered for publication in Nutrients and we are looking forward to hearing good news.

Sincerely,

Hyo-Joon Yang, MD, PhD

Associate Professor

Division of Gastroenterology, Department of Internal Medicine and Gastrointestinal Cancer Center, Kangbuk Samsung Hospital, Sungkyunkwan University School of Medicine, Seoul 03181, Korea.

Tel: 82-2-2001-8330, Fax: 82-2-2001-8360, E-mail: [email protected]

Point-by-point response

Response to the section editor’s comment

Comments for the Author
We found that your paper's repetition rate is 39%. And the repetition rate of your paper is 6% with one paper Based on our rules, the repetition rate of your paper can not exceed 30%, and the repetition rate can not be more than 10% with one paper. Please modify your paper and make sure that the repetition rate less than 30%, and less than 10% with one same paper.

Response: We would like to thank to the section editor’s comment and apologize for this high repetition rate of the original manuscript. We revised the manuscript so than the repetition rate is 29% after excluding quote and bibliography. We attached the ithenticate report.

Response to the reviewers’ comments

Reviewer 1

Comments for the Author
The authors investigated the role of the microbiota in H.pylori negative CG patients that went on to develop GC. While the paper is well written, there were some major problems with the presentation and thus interpretation of the results. Once important changes are made and the results are still significant, the paper could be publishable.

  1. The authors investigated several cytokines as well as several bacterial species, but no mention of multiple testing correction is made in the entire paper. This must be changed for the correlations as well as the boxplots shown in the supplementary material. Without correction, the authors cannot be confident that the results are not spurious associations.

Response: We express our gratitude for the reviewer’s comment. In the microbial profiling using analysis of comparison of microbiomes (ANCOM) 2, we used the taxa-wise false discovery rate (FDR) adjusted p value using Benjamini-Hochberg adjustment for the multiple testing correction method. An FDR significance threshold of 0.1 was chosen for calculation of W statistics. Considering the fact that FDR<0.1 is also widely utilized in the field of biomedical science [1], as well as difference abundance analysis using ANCOM [2], we chose to use FDR<0.1 as the cutoff in this analysis. Additionally, the statistical decision made by ANCOM depends on the quantile of its test statistic W, rather than pvalues. For each taxon, the number of rejections, denoted by Wi, is counted, and ANCOM makes use of the empirical distribution of (W1, W2, …, Wm) to determine the cut-off value of significant taxon. The rule of thumb is that the larger the value of Wi is, the more likely the taxon i will be differentially abundant. Although the ANCOM outputs results from different cutoffs such as the 60th to 90th percentile, W statistics ≥70% of the total number of genera tested were considered significant because the authors of the method recommended the cutoff level [3], and we showed the detection levels (≥70%) in Table 2. We did not use FDR correction for multivariate association with linear models (MaAsLin) because this method was used only for the validation of the significant findings in the analysis using ANCOM. Although there were some significant findings (p < 0.05) in the MaAsLin analysis that were not significant in the ANCOM analysis, we did not describe those results. The boxplot shown in Figure S4 was intended to show the relative abundance of the significant taxa listed in Table 2. We had already mentioned it in the Method section (Lines: 147–150) and the footnote of the Table 2, but we have revised the Methods section more clearly (Lines: 144–152 and Table 2). We also revised the legend of Figure S4.

We made the following change in this revision:

  1. Methods (lines 144–152)

Initial sentences: An ANCOM2 detection level ≥0.7 was considered significant as it indicates that the ratios of the taxon to at least 70% of other taxa were detected to be significantly different (FDR, q < 0.10) between the cancer and control groups. The significance threshold of the detection level of 0.7 was recommended by the author of ANCOM2 as a common choice [19]. For MaAsLin2 models, the control group was used as the reference group

Revised sentences: The taxa-wise false discovery rate (FDR) adjusted p value using Benjamini-Hochberg adjustment for the multiple testing correction method was used for the ANCOM2. An ANCOM2 detection level ≥ 0.7 was considered significant as it indicates that the ratios of the taxon to at least 70% of other taxa were detected to be significantly different (FDR, q < 0.10) between the cancer and control groups. The significance threshold of the de-tection level of 0.7 was recommended by the author of ANCOM2 as a common choice [19]. For MaAsLin2 models, the control group was used as the reference group, and the FDR correction was not used because these models were used only for the validation of the significant findings in the analysis using ANCOM2.

  1. Table 2 footnote

Initial sentences: Taxa detected to be significantly different relative to more than 70% of other taxa at each taxon level (detected > 0.7) in the analysis of comparison of microbiomes (ANCOM)-2 analysis are in-cluded in this table. aNCBI-RefSeq database was used for taxonomic assignment. bBenjamini–Hochberg multiple comparison correction method was applied (false discovery rate < 0.1). cCoef-ficients for log-transformed relative abundance of each taxon in linear models using MaAsLin2. The control group was set as the reference group and compared to the cancer group. The significance of the ANCOM2 model was validated using the MaAsLin2 model. ANCOM, analysis of comparison of microbiomes; MsAsLin, multivariate association with linear models; BMI, body mass index.

Revised sentences: Taxa detected to be significantly different relative to more than 70% of other taxa at each taxon level (detected > 0.7) in the analysis of comparison of microbiomes (ANCOM)-2 analysis are in-cluded in this table. aNCBI-RefSeq database was used for taxonomic assignment. bBenjamini–Hochberg multiple comparison correction method was applied (false discovery rate < 0.1). cCoef-ficients for log-transformed relative abundance of each taxon in linear models using MaAsLin2. The control group was set as the reference group and compared to the cancer group. dUnadjusted p values for the multiple comparison. The significance of the ANCOM2 model was validated using the MaAsLin2 model. ANCOM, analysis of comparison of microbiomes; MsAsLin, multivariate association with linear models; BMI, body mass index; FC, fold change.

  1. Figure S4 legend (supplement file)

Initial sentences: Figure S4. Box plots showing the relative abundance of taxa that differed significantly between the control and cancer groups.

Revised sentences: Figure S4. Box plots showing the relative abundance of taxa that differed significantly between the control and cancer groups in Table 2. Taxa detected to be significantly different relative to more than 70% of other taxa at each taxon level (detected > 0.7) in the analysis of comparison of microbiomes (ANCOM)-2 analysis (false discovery rate [FDR] < 0.1).

The results of the cytokine and correlation analysis may have been the most important reason for the reviewer to make this comment. We agree with the reviewer that the correction for multiple comparisons helps avoiding false positive findings. However, we believe that the p value is also meaningful for the cytokine analysis so that the readers can compare the current results with those of other studies because results of cytokine analysis are not usually corrected for multiple comparison in many studies [4-7]. Therefore, we presented both p value and FDR-corrected q value. In results, the higher relative mRNA expression level of IL-1β in the gastric mucosa in the cancer group compared to the control group was not significant after the FDR correction (q = 0.210). The three-group comparison among chronic gastritis (CG), intestinal metaplasia (IM), and gastric cancer (GC) showed similar results (q = 0.063). We also applied the FDR adjustment for the correlation analysis between the microbial profiles and the relative mRNA expression of IL-1β. We also removed the correlation analysis between the microbial profiles and the relative mRNA expression of the other cytokines including Table S2 of the original manuscript to minimize the risk of false positive findings. We revised the Methods section (lines, 187–189) and the Results section (lines, 317–325 and 337–349).

We made the following change in this revision:

  1. Methods (lines 187–189):

Initial sentences: Statistical analyses of cytokine expression and correlation analyses were performed using R.

Revised sentences: Statistical analyses of cytokine expression and correlation analyses were performed using R. The FDR-corrected q value was also used for the results of mRNA expression and correlation analyses.

  1. Results (lines 317–325):

Initial sentences: In RT-qPCR analysis, the relative mRNA expression level of IL-1β in the gastric mucosa was significantly higher in the cancer group than that in the control group (Mann-Whitney test, p = 0.030) (Figure 6A). Although the mean expression levels of IL-6, IL-8, IL-10, and IL-17A were higher in the cancer group than those in the control group, the difference was not statistically significant. The mRNA expression of IL-1β also showed a significantly increasing trend in CG, IM, and GC in the three-group comparison (Linear regression analysis, p-for-trend = 0.009) (Figure 6B).

Revised sentences: In RT-qPCR analysis, the relative mRNA expression level of IL-1β in the gastric mucosa was significantly higher in the cancer group than that in the control group (Mann-Whitney test, p = 0.030) (Figure 6A). However, this was not significant after the correction for multiple comparison (q = 0.210). Although the mean expression levels of IL-6, IL-8, IL-10, and IL-17A were higher in the cancer group than those in the control group, the difference was not statistically significant. The mRNA expression of IL-1β also showed a significantly increasing trend in CG, IM, and GC in the three-group comparison, but also was not significant after the FDR correction (Linear regression analysis, p-for-trend = 0.009, q = 0.063) (Figure 6B).

  1. Results (lines 337–349):

Initial sentences: The α-diversity indices that were reduced in GC were negatively correlated with the mRNA expression of IL-1β (observed, p = 0.002, r = −0.4381; Fisher, p = 0.002, r = −0.4471) (Figure 7A). We analyzed the correlation between the microbial profile associated with GC and pro-inflammatory cytokine expression. The α-diversity indices that were reduced in GC were negatively correlated with the mRNA expression of IL-1β (observed, p = 0.002, r = −0.4381; Fisher, p = 0.002, r = −0.4471) (Figure 7A). These indices were also negatively correlated with the mRNA expression levels of IL-6, IL-8, and IL-17A (all p < 0.05) (Table S2). Among the differentially associated taxa between GC and control groups, the rela-tive abundance of Campylobacter was negatively correlated with IL-1β (p = 0.003, r = −0.4328) and IL-6 (p < 0.05) mRNA expression levels (Figure 7B; Table S2). Although IL-1β mRNA expression correlated positively with Lacticaseibacillus and negatively with Haemophilus, these correlations were not statistically significant. As the genera Campylobacter, Lacticaseibacillus, and Haemophilus were the three most important taxa in the RF model, we combined the relative abundance of these taxa to create a GC microbiome index. The GC microbiome index was significantly positively correlated with IL-1β mRNA expression (p = 0.031, r = 0.3185).

Revised sentences: After excluding outliers with values outside the mean ± 2 standard deviations, the observed ASV that was reduced in GC was negatively correlated with the mRNA ex-pression of IL-1β (q = 0.033, r = −0.3260) (Figure 7A). However, when we further excluded two more highest value points of the observed ASV, the correlation became no longer significant (q = 0.214, r = −0.2267), suggesting that this correlation may have been driven by several high values of the α-diversity. Among the differentially associated taxa be-tween GC and control groups, the relative abundance of Campylobacter was negatively correlated with IL-1β (q = 0.008, r = −0.4328) (Figure S6). However, because Campylobacter was absent in many samples, this correlation might have driven by several samples with high abundance of this taxon. As the genera Campylobacter, Lacticaseibacillus, and Haemophilus were the three most important taxa in the RF model, we combined the relative abundance of these taxa to create a GC microbiome index. The GC microbiome index was significantly positively correlated with IL-1β mRNA expression (q = 0.031, r = 0.3185).

  1. Figure 4 has very little meaning, as the samples were not allowed to cluster. There is no observable pattern in the heatmap. The authors should consider coloring the column groups by the three groups mentioned in the caption of the figure and to allow the samples to cluster to identify any observable clusters in the data.

Response: As we showed the differentially abundant taxa in Table 2, we found only three genus (Campylobacter, Haemophilus, and Lacticaseibacillus) were significantly different between the control and cancer groups. Therefore, we could not find distinctly clustered pattern in the heatmap including all genera. Nevertheless, we believed the heatmap was better than bar plots to show the relative abundance of all species in our total samples. As the reviewer’s advice, we revised the Results section and also colored the column groups by the three groups of CG, IM, and GC in Figure 4.

We made the following change in this revision:

  1. Results (lines 265–268):

Initial sentences: We also observed a higher abundance of Lacticaseibacillus and a lower abundance of Haemophilus in samples with GC compared to the control samples in the hierarchical clustering heatmap (Figure 4).

Revised sentences: We also observed a higher abundance of Lacticaseibacillus and a lower abundance of Campylobacter and Haemophilus in samples with GC compared to the control samples in the hierarchical clustering heatmap although the heatmaps were not distinctly clustered by the two groups (Figure 4).

  1. Figure 4 legend

Initial sentences: Differentially abundant gastric microbiota between cancer and control groups at the genus level in the hierarchical clustering heatmap. Two-colored bars on the top of the figure show the grouping category. Every sample (control group = red, cancer group = green) is represented by its own column. Scaling is by relative abundances of each taxon from low (light blue) to high (dark red) across all samples. Clustering was performed using Euclidean distance measure and Ward’s linkage clustering al-gorithm at the genus level. Individual patient identifiers are listed followed by CG, IM, or GC, indicating chronic gastritis, intestinal metaplasia, or gastric cancer, respectively.

Revised sentences: Hierarchical clustering heatmap for relative abundance of gastric micro-biota between cancer and control groups (Group 1) at the genus level. Two-colored bars on the top of the figure show the grouping category. Every sample (control group = red, cancer group = green in Group 1) is represented by its own column. Scaling is by relative abundances of each taxon from low (light blue) to high (dark red) across all samples. Clustering was performed using Euclidean distance measure and Ward’s linkage clustering al-gorithm at the genus level. Individual patient identifiers are listed followed by CG, IM, or GC (Group 2), indicating chronic gastritis, intestinal metaplasia, or gastric cancer, respectively.

  1. Correlation plots are important for the understanding of the paper. In Figure 7A the 4 samples on the right, i.e. the high end of diversity, are driving the correlations. It also appears to be the same 4 samples in both plots. Please show the correlations without these 4 extreme samples.

Response: We agree with the reviewer that the outliers should be removed in a correlation analysis. When we removed the four samples with highest values of observed index, there were no longer significant association between this index and the mRNA expression of IL-1β (q = 0.214, r = −0.2267). However, because it was unclear whether these four values are outliers or not, we further analyzed the correlation with setting outliers as values outside the mean ± 2 standard deviations (SDs) (5 percentile of the values). Because the mean ± SD of observed index was 47.2 ± 32.9, observed index > 113 was considered outliers and removed from the analysis. In the results, the correlation was significant after correcting for multiple comparison (q = 0.033, r = −0.3260) between observed index and the relative mRNA expression of IL-1β. Therefore, we would like to suggest that, although there is a significant correlation even after removing outliers and correcting multiple comparison, this correlation may have been driven by several high values of the observed index. We revised the Results section (lines 337–342) and replaced Figure 7A with Figure R2.

We made the following change in the Results section (lines 337–342):

Initial sentences: The α-diversity indices that were reduced in GC were negatively correlated with the mRNA expression of IL-1β (observed, p = 0.002, r = −0.4381; Fisher, p = 0.002, r = −0.4471) (Figure 7A).

Revised sentences: After excluding outliers with values outside the mean ± 2 standard deviations, the observed index that was reduced in GC was negatively correlated with the mRNA expression of IL-1β (q = 0.033, r = −0.3260) (Figure 7A). However, when we further excluded two more highest value points of the observed index, the correlation became no longer significant (q = 0.214, r = −0.2267), suggesting that this correlation may have been driven by several high values of the α-diversity.

  1. Further, why the authors are showing two measures of the same diversity when it appears to be just a different unit is a mystery. The authors should only use one of the measures and put the other into the supplementary.

Response: We used the several indices to compare the α-diversity calculated from indices where various definitions might be influenced by different assumptions of species diversity. Observed feature is simply the number of observed ASVs, while Fisher’s alpha index accounts for relationship between the number of species and the abundance of each species and Shannon’s index accounts for both abundance and evenness of the taxa present. Nevertheless, according to the reviewer’s comment, we removed the plot of Fisher’s index in Figure 2 and maintained the observed ASV and Shannon’s index to account for richness and both richness and evenness, respectively(Lines 116–118 and 209–212). As a result, we also removed the correlation between the Fisher’s index and the relative mRNA expression of IL-1β in Fig 7A.

We made the following change in this revision:

  1. Methods (lines 116–118):

Initial sentences: We calculated richness using the actual number of unique taxa observed in each sample (observed index) and used Shannon and Fisher indices to describe both richness and evenness.

Revised sentences: We calculated richness using the actual number of unique taxa observed in each sample (observed index) and used Shannon index to describe both richness and evenness.

  1. Results (lines 209–212):

Initial sentences: Considering both richness and evenness, we found significantly lower α-diversity using the Fisher index in patients with GC (p = 0.011; Figure 2B), while there was no difference in the Shannon index between the two groups (p = 0.090; Figure 2C). These results were consistent after additional adjustments for age, sex, and BMI (analysis of covariance, p = 0.047, p = 0.048, and p = 0.898, respectively).

Revised sentences: Considering both richness and evenness, there was no difference in the Shannon index between the two groups (p = 0.090; Figure 2B). These results were consistent after additional adjustments for age, sex, and BMI (analysis of covariance, p = 0.047 and p = 0.898, respectively).

  1. Fig 2 legend:

Initial sentences: Figure 2. Reduced α-diversity in the gastric microbiota of patients with cancer than that in the controls. Boxplots represent the α-diversity measured using (A) observed amplicon sequence variants (ASVs), (B) Fisher index, and (C) Shannon index at ASV level.

Revised sentences: Figure 2. Reduced α-diversity in the gastric microbiota of patients with cancer than that in the controls. Boxplots represent the α-diversity measured using (A) observed amplicon sequence variants (ASVs) and (B) Shannon index at ASV level.

  1. Results (lines 337–339):

Initial sentences: The α-diversity indices that were reduced in GC were negatively correlated with the mRNA expression of IL-1β (observed, p = 0.002, r = −0.4381; Fisher, p = 0.002, r = −0.4471) (Figure 7A).

Revised sentences: After excluding outliers with values outside the mean ± 2 standard deviations, the observed index that was reduced in GC was negatively correlated with the mRNA expression of IL-1β (q = 0.033, r = −0.3260) (Figure 7A).

  1. The first 3 panels of Figure 7B are between worrying and not true correlations, as the correlations are driven again by extreme values in those patients where the bacteria were present. Here, the authors should either not show those samples where the bacteria were absent or choose a different mode of association. A fisher test could be recommended here where the presence and absence of the bacteria are coded as 1 and 0, respectively. The only panel that showed a true correlation is the last panel of Figure 7B but since there was no multiple testing correction applied, it cannot be trusted.

Response: We performed both analyses the reviewer recommended. First, when we removed samples where the bacteria were absent, the relative abundance of Campylobacter was still negatively correlated with IL-1β (p = 0.033, r = −0.5780), but this was not significant after the FDR correction (q = 0.100). However, this method may be limited because they do not reflect the increased relative mRNA expression of IL-1β in the absence of Campylobacter. Second, we categorized the absence and presence of each taxon as 0 and 1, respectively. Because the Fisher’s exact test can compare two categorical variables, we also categorized the decrease and increase of relative mRNA expression of IL-1β as 0 and 1, respectively. There were no significant associations in this analysis (p > 0.05). However, this method may also be limited because of substantial information loss during categorization. Third, we maintained the categorization of the absence and presence of each taxon as 0 and 1, respectively, and compared the relative mRNA expression of IL-1β between the two categories using the Mann-Whitney U test. The relative mRNA expression of IL-1β in samples without Campylobacter was significantly higher than those in the samples with Campylobacter (p = 0.018) and marginally significant after the FDR correction (q = 0.053). Because all of these methods had limitations, we would like to suggest maintaining the current version of correlation because this is the same method used for the correlation between the GC microbiome index and the relative mRNA expression of IL-1β. However, according to the reviewer’s concern, we changed the p value into the FDR-corrected q value and moved to supplementary file (Figure S6). We also commented that because Campylobacter was absent in many samples, this correlation might have driven by several samples with high abundance of this taxon. We believe this limitation supports the need for generating GC microbiome index to evaluate the correlation with the pro-inflammatory cytokine expression.

We made the following change in the Results section (lines 342–346):

Initial sentences: Among the differentially associated taxa between GC and control groups, the relative abundance of Campylobacter was negatively correlated with IL-1β (p = 0.003, r = −0.4328) and IL-6 (p < 0.05) mRNA expression levels (Figure 7B; Table S2). Although IL-1β mRNA expression correlated positively with Lacticaseibacillus and negatively with Haemophilus, these correlations were not statistically significant.

Revised sentences: Among the differentially associated taxa between GC and control groups, the relative abundance of Campylobacter was negatively correlated with IL-1β (q = 0.0083, r = −0.4328) (Figure S6). However, because Campylobacter was absent in many samples, this correlation might have driven by several samples with high abundance of this taxon.

Reviewer 2

Comments for the Author
I have had the opportunity of reading the paper from Kim et al. This is one of the first studies to describe the interactions between gastric microbiota and ongoing inflammation in H. pylori-negative patients. The manuscript is easy to understand and presents data fairly with an attempt to explain adjusted models as well as the limitations of the data.

  1. The 16S rRNA gene sequencing and cytokine mRNA expression have been done with 67 and 113 patients, respectively, meanwhile the microbiome-cytokine association analysis has done with only 47 patients. Are the microbiota composition and cytokine expression from these patients representative? Please, include these results in supplementary data.

Response: We appreciate the reviewer’s comment. We evaluated the representativeness of the 47 patients for the overall study poopulation in three levels. First, in the patient demographics, the 47 patients showed similar clinical characteristics with the overall population (Table S2). Second, the relative abudnance of the differentially abundant taxa were similar between the overall patients and the patients included in the correlation analysis (Table S3). There was no significant results in the ANCOM and MaAsLin analyses in this population because of small sample size. However, the directions of association that were presented as coefficients for log-transformed relative abundance (log2 fold change [FC]) were consistent with those in the original population after adjustment for age, sex, and body mass index. Third, the relative mRNA expression level of IL-1β in the gastric mucosa was also significantly higher in the cancer group than in the control group among these patients (Mann-Whitney test, p = 0.015, q = 0.105) (Figure S5A). In addition, The mRNA expression of IL-1β also showed a significantly increasing trend in CG, IM, and GC in the three-group comparison of these patients (linear regression analysis, p-for-trend = 0.008, q = 0.056) (Figure S5B). We revised the Results (lines 333–337) and added Table S2, Table S3, and Figure S5 in the supplement file.

We made the following change in the Results section (lines 333–337):

Initial sentences: We analyzed the correlation between the microbial profile associated with GC and pro-inflammatory cytokine expression.

Revised sentences: We analyzed the correlation between the microbial profile associated with GC and pro-inflammatory cytokine expression. The patients included in this analysis showed similar baseline characteristics (Table S2), relative abundance of the differentially abundant taxa (Table S3), and cytokine expression profile (Figure S2) compared to the overall population.

Reviewer 3

Comments for the Author
In this manuscript, the authors aimed to determine the relationship of different gastric mucosal pathological status and microbiome in Helicobacter pylori-negative gastric cancer. It is an interesting story. But the data interpretation is confusing in several places and the authors need to provide a better description/discussion of the manuscript. Therefore, more work is needed to address in depth in order to meet the standards of the journal. A major revision is recommended.

  1. In Pubmed database, several papers had revealed the role of non-Helicobacter pylori bacteria in the pathogenesis of gastroduodenal diseases. Would you like to introduce more detail of your research to increase the novelity?

Response: We thank the reviewer for the reviewer’s comments. The novelty of our study is that we identified gastric microbiota associated with GC in H. pylori-negative patients, which has growing importance. Previous studies have focused on the synergistic role of H. pylori and other gastric microbiota in GC [8-11]. Sung et al. reported gastric microbiota associated with the progression of the gastric premalignant lesion after H. pylori eradication [12]. However, studies specifically investigating gastric microbiota associated with GC in H. pylori-negative patients have been limited. We adopted strict criteria for classifying H. pylori-negative cases using 16S rRNA gene sequencing data, which allowed us to evaluate the gastric microbial community, excluding H. pylori [13]. Another novelty is that we investigated the gastric mucosal inflammatory cytokine response associated with differentially abundant gastric microbiota in GC. Recently, several studies reported the inflammatory cytokine response to the gastrointestinal microbiota in healthy control or patients with Parkinson’s disease [7,14]. The role of IL-1β, IL-6, and IL-8 has been noted in the association between chronic H. pylori infection and the development of GC [15]. However, investigations that took the interactions between ongoing inflammation and the microbiome into account have been limited. To our knowledge, this is the first study that evaluated the gastric microbiota and inflammatory cytokine response in association with GC in H. Pylori-negative GC. We added these points in the Discussion section (lines, 431–444).

We made the following change in the Discussion section (lines 431–444):

Initial sentences: The strength of our study is the strict criteria for classifying H. pylori-negative cases using 16S rRNA gene sequencing data, which allowed us to evaluate the gastric microbial community, excluding H. pylori.

Revised sentences: The novelty of our study is that we identified gastric microbiota associated with GC in H. pylori-negative patients, which has growing importance. Previous studies have focused on the synergistic role of H. pylori and other gastric microbiota in GC [8–10,21]. Sung et al. reported gastric microbiota associated with the gastric premalignant lesion after H. pylori eradication [26]. We adopted strict criteria for classifying H. pylori-negative cases using 16S rRNA gene sequencing data, which allowed us to evaluate the gastric microbial community, excluding H. pylori [20]. Another novelty is that we investigated the gastric mucosal inflammatory cytokine response associated with differentially abundant gastric microbiota in GC. Recently, several studies reported the inflammatory cytokine response to the gastrointestinal microbiota in healthy control or patients with Parkinson’s disease [37,38]. The role of IL-1β, IL-6, and IL-8 has been noted in the association between chronic H. pylori infection and the development of GC [14]. To our knowledge, this is the first study that evaluated the gastric microbiota and inflammatory cytokine response in association with H. pylori-negative GC.

  1. Most of cytokines are soluble or, secretable. But in your research, you just focus on the colonizing bacteria in gastric mucosal. Would you ever consider both tissue and gastric juice? Especially the cytokines in gastric juice.

Response: We agree with the reviewer that the direct measurement of cytokine levels could have provided more robust results. Several studies reported plasma concentration levels of cyokines [6,7]. However, we would like to suggest that the measurement of mRNA expression levels of inflammatory cytokine genes from gastric mucosal samples also can provide reliable results [16,17]. We acknowledged this point in the limitaton of the Dicussion section (lines, 446–449).

We added the following sentence in the Discussion section (lines 446–449):

Second, we did not measure cytokine levels from gastric juice or plasma. However, the measurement of mRNA expression levels of cytokine genes from gastric mucosal samples may also provide meaningful results [39,40].

  1. The procedure of endoscopy should be discribed detailedly, because the operation steps is vital for the following experiments.

Response: An experienced gastrointestinal endoscopist performed the procedure using a single-channel endoscope (GIF-H290; Olympus Optical, Tokyo, Japan). Before endoscopy, patients fasted overnight and rinsed their mouths. To avoid contamination, a disinfected endoscope was used according to the standard sterilization protocol. In addition, endoscopic biopsy samples for 16S rRNA gene analysis were obtained before any fluid was suctioned through the scope. Endoscopic procedure included detailed observation of esophageal, gastric, and duodenal lesions. Gastric mucosal atrophy was evaluated endoscopically. Gastric intestinal metaplasia (IM) was evaluated by histological evaluation using endoscopic biopsy tissues taken from both the lesser curvature side of the mid-antrum and the lesser curvature side of the mid-body of the stomach. H. pylori infection was evaluated by histological evaluation with modified Giemsa staining and rapid urease test and was considered negative when both tests were negative. We revised the Methods section to augment the endoscopy procedure (lines, 72–83).

We made the following change in the Methods section (lines 72–83):

Initial sentences: To avoid contamination, a disinfected endoscope was used according to the standard sterilization protocol [15,16]. Patients fasted overnight and rinsed their mouths before endoscopy. Endoscopic biopsy samples were obtained before any fluid was suctioned through the scope. Gastric IM was evaluated by histological evaluation using endoscopic biopsy tissues taken from both the lesser curvature side of the mid-antrum and the lesser curvature side of the mid-body of the stomach. H. pylori infection was evaluated by histological evaluation with modified Giemsa staining and rapid urease test and was considered negative when both tests were negative.

Revised sentences: To avoid contamination, a disinfected endoscope was used according to the standard sterilization protocol [15,16]. Patients fasted overnight and rinsed their mouths before endoscopy. An experienced gastrointestinal endoscopist performed the procedure using a single-channel endoscope (GIF-H290; Olympus Optical, Tokyo, Japan). Endoscopic biopsy samples were obtained before any fluid was suctioned through the scope. Endoscopic procedure included detailed observation of esophageal, gastric, and duodenal lesions. Gastric mucosal atrophy was evaluated endoscopically. Gastric IM was evaluated by histological evaluation using endoscopic biopsy tissues taken from both the lesser curvature side of the mid-antrum and the lesser curvature side of the mid-body of the stomach. H. pylori infection was evaluated by histological evaluation with modified Giemsa staining and rapid urease test and was considered negative when both tests were negative.

  1. The baseline demographics is too simple, please provide more characteristics of the patients.

Response: We evaluated gastric mucosal atrophy and intestinal metaplasia during endoscopy. Atrophy was evalauted endoscopically, and IM was evaluated histologically. As expected, the proportions of patients with gastric mucosal atrophy and IM were significantly higher in the cancer group than in the control group (both p <0.001). We added these baseline characteristics in the Table 1.

  1. In Fig.4, the data of normal, intestinal metaplasia, and cancer should be shown because they are a continuous disease spectrum.

Response: As the reviewer’s advice, we added a colored bar grouped by the three groups of CG, IM, and GC in Figure 4.

References

  1. Jiang, L.; Amir, A.; Morton, J.T.; Heller, R.; Arias-Castro, E.; Knight, R. Discrete False-Discovery Rate Improves Identification of Differentially Abundant Microbes. mSystems 2017, 2, doi:10.1128/mSystems.00092-17.
  2. Plachokova, A.S.; Andreu-Sanchez, S.; Noz, M.P.; Fu, J.; Riksen, N.P. Oral Microbiome in Relation to Periodontitis Severity and Systemic Inflammation. Int J Mol Sci 2021, 22, doi:10.3390/ijms22115876.
  3. Lin, H.; Peddada, S.D. Analysis of microbial compositions: a review of normalization and differential abundance analysis. NPJ Biofilms Microbiomes 2020, 6, 60, doi:10.1038/s41522-020-00160-w.
  4. Kaur, R.P.; Vasudeva, K.; Singla, H.; Benipal, R.P.S.; Khetarpal, P.; Munshi, A. Analysis of pro- and anti-inflammatory cytokine gene variants and serum cytokine levels as prognostic markers in breast cancer. J Cell Physiol 2018, 233, 9716-9723, doi:10.1002/jcp.26901.
  5. Li, L.; Chen, L.; Zhang, W.; Liao, Y.; Chen, J.; Shi, Y.; Luo, S. Serum cytokine profile in patients with breast cancer. Cytokine 2017, 89, 173-178, doi:10.1016/j.cyto.2015.12.017.
  6. Milic, L.; Karamarkovic, A.; Popadic, D.; Sijacki, A.; Grigorov, I.; Milosevic, E.; Cuk, V.; Pesko, P. Altered cytokine expression in Helicobacter pylori infected patients with bleeding duodenal ulcer. BMC Res Notes 2019, 12, 278, doi:10.1186/s13104-019-4310-4.
  7. Lin, C.H.; Chen, C.C.; Chiang, H.L.; Liou, J.M.; Chang, C.M.; Lu, T.P.; Chuang, E.Y.; Tai, Y.C.; Cheng, C.; Lin, H.Y., et al. Altered gut microbiota and inflammatory cytokine responses in patients with Parkinson's disease. J Neuroinflammation 2019, 16, 129, doi:10.1186/s12974-019-1528-y.
  8. Wang, Z.; Gao, X.; Zeng, R.; Wu, Q.; Sun, H.; Wu, W.; Zhang, X.; Sun, G.; Yan, B.; Wu, L., et al. Changes of the Gastric Mucosal Microbiome Associated With Histological Stages of Gastric Carcinogenesis. Front Microbiol 2020, 11, 997, doi:10.3389/fmicb.2020.00997.
  9. Gantuya, B.; El Serag, H.B.; Matsumoto, T.; Ajami, N.J.; Uchida, T.; Oyuntsetseg, K.; Bolor, D.; Yamaoka, Y. Gastric mucosal microbiota in a Mongolian population with gastric cancer and precursor conditions. Aliment Pharmacol Ther 2020, 51, 770-780, doi:10.1111/apt.15675.
  10. Ferreira, R.M.; Pereira-Marques, J.; Pinto-Ribeiro, I.; Costa, J.L.; Carneiro, F.; Machado, J.C.; Figueiredo, C. Gastric microbial community profiling reveals a dysbiotic cancer-associated microbiota. Gut 2018, 67, 226-236, doi:10.1136/gutjnl-2017-314205.
  11. Coker, O.O.; Dai, Z.; Nie, Y.; Zhao, G.; Cao, L.; Nakatsu, G.; Wu, W.K.; Wong, S.H.; Chen, Z.; Sung, J.J.Y., et al. Mucosal microbiome dysbiosis in gastric carcinogenesis. Gut 2018, 67, 1024-1032, doi:10.1136/gutjnl-2017-314281.
  12. Sung, J.J.Y.; Coker, O.O.; Chu, E.; Szeto, C.H.; Luk, S.T.Y.; Lau, H.C.H.; Yu, J. Gastric microbes associated with gastric inflammation, atrophy and intestinal metaplasia 1 year after Helicobacter pylori eradication. Gut 2020, 69, 1572-1580, doi:10.1136/gutjnl-2019-319826.
  13. Kim, H.N.; Kim, J.H.; Chang, Y.; Yang, D.; Kim, H.L.; Ryu, S. Gut Microbiota Composition across Normal Range Prostate-Specific Antigen Levels. J Pers Med 2021, 11, doi:10.3390/jpm11121381.
  14. Schirmer, M.; Smeekens, S.P.; Vlamakis, H.; Jaeger, M.; Oosting, M.; Franzosa, E.A.; Ter Horst, R.; Jansen, T.; Jacobs, L.; Bonder, M.J., et al. Linking the Human Gut Microbiome to Inflammatory Cytokine Production Capacity. Cell 2016, 167, 1125-1136 e1128, doi:10.1016/j.cell.2016.10.020.
  15. Tsujimoto, H.; Ono, S.; Ichikura, T.; Matsumoto, Y.; Yamamoto, J.; Hase, K. Roles of inflammatory cytokines in the progression of gastric cancer: friends or foes? Gastric cancer : official journal of the International Gastric Cancer Association and the Japanese Gastric Cancer Association 2010, 13, 212-221, doi:10.1007/s10120-010-0568-x.
  16. Outlioua, A.; Badre, W.; Desterke, C.; Echarki, Z.; El Hammani, N.; Rabhi, M.; Riyad, M.; Karkouri, M.; Arnoult, D.; Khalil, A., et al. Gastric IL-1beta, IL-8, and IL-17A expression in Moroccan patients infected with Helicobacter pylori may be a predictive signature of severe pathological stages. Cytokine 2020, 126, 154893, doi:10.1016/j.cyto.2019.154893.
  17. Kim, H.J.; Kim, N.; Park, J.H.; Choi, S.; Shin, C.M.; Lee, O.J. Helicobacter pylori Eradication Induced Constant Decrease in Interleukin- 1B Expression over More Than 5 Years in Patients with Gastric Cancer and Dysplasia. Gut and liver 2020, 14, 735-745, doi:10.5009/gnl19312.

Reviewer 2 Report

I have had the opportunity of reading the paper from Kim et al. This is one of the first studies to describe the interactions between gastric microbiota and ongoing inflammation in H. pylori-negative patients. The manuscript is easy to understand and presents data fairly with an attempt to explain adjusted models as well as the limitations of the data.

Comments:

-         The 16S rRNA gene sequencing and cytokine mRNA expression have been done with 67 and 113 patients, respectively, meanwhile the microbiome-cytokine association analysis has done with only 47 patients. Are the microbiota composition and cytokine expression from these patients representative? Please, include these results in supplementary data.

Author Response

(The authors gave the same response as above.)

Reviewer 3 Report

In this manuscript, the authors aimed to determine the relationship of different gastric mucosal pathological status and microbiome in Helicobacter pylori-negative gastric cancer. It is an interesting story. But the data interpretation is confusing in several places and the authors need to provide a better description/discussion of the manuscript. Therefore, more work is needed to address in depth in order to meet the standards of the journal. A major revision is recommended.

1. In Pubmed database, several papers had revealed the role of non-Helicobacter pylori bacteria in the pathogenesis of gastroduodenal diseases.  Would you like to introduce more detail of your research to increase the novelity?

2. Most of cytokines are soluble or, secretable. But in your research, you just focus on the colonizing bacteria in gastric mucosal. Would you ever consider both tissue and gastric juice? Especially the cytokines in gastric juice.

3.The procedure of endoscopy should be discribed detailedly, because the operation steps is vital for the following experiments.

4.The baseline demographics is too simple, please provide more characteristics of the patients.

5. In Fig.4, the data of normal, intestinal metaplasia, and cancer should be shown because they are a continuous disease spectrum.

Author Response

(The authors gave the same response as above.)

Round 2

Reviewer 1 Report

Dear authors, 

thank you for addressing my concerns. My only one concern is now to pick one FDR threshold and stick with it. In the manuscript you vary between what is significant: an FDR < .1 or <.05? please pick one. 

Other than that, the manuscript is greatly improved.